# Public Facility Utility and Third-Hand Smoking Exposure without First and Second-Hand Smoking According to Urinary Cotinine Level

**DOI:** 10.3390/ijerph16050855

**Published:** 2019-03-08

**Authors:** Si Yun Moon, Tae Won Kim, Yoon-Ji Kim, Youngki Kim, Se Yeong Kim, Dongmug Kang

**Affiliations:** 1Department of Premedicine, School of Medicine, Pusan National University, Yangsan, Gyongnam 50612, Korea; siyoun806@pusan.ac.kr (S.Y.M.); xodnjs1390@pusan.ac.kr (T.W.K.); 2Department of Preventive, and Occupational & Environmental Medicine, School of Medicine, Pusan National University, Yangsan, Gyongnam 50612, Korea; harrypotter79@pusan.ac.kr (Y.-J.K.); mungis@pusan.ac.kr (Y.K.); 3Department of Occupational and Environmental Medicine, Pusan National University Yangsan Hospital, Yangsan, Gyongnam 50612, Korea; 30white@pusan.ac.kr

**Keywords:** third hand smoke, tobacco, transportation, susceptible, environmental health

## Abstract

Third-hand smoke (THS) causes pathological changes in the liver, lungs, and skin. THS exposure can be ubiquitous, chronic, and unconscious. However, little is known about THS exposure in public facilities and its susceptible population. This paper aimed to identify which public facilities and socio-demographic groups were especially vulnerable to THS. Data from 1360 adults obtained from Korean National Environmental Health Survey I (2009–2011) were analyzed. To study the sole effect of THS, we restricted the study population to those participants who had never smoked and who had no exposure to second-hand smoke. The assessed variables included the type and frequency of public transportation, frequency of use of 12 different public facilities, and 8 socio-demographic factors. Urinary cotinine was used as a biomarker. *T*-tests and analysis of variance were used for univariate analyses, while generalized linear regression was used for multivariate analysis. Frequent use of public transportation, bars, internet cafés, and participants with low levels of education, divorced or bereaved, living in multi-unit houses, and with smokers within the family were associated with significantly high urinary cotinine levels. These findings indicate that the frequent use of public transportation, certain public facilities and certain socio-demographic factors can result in high THS exposure.

## 1. Introduction

First-hand smoke (FHS) and second-hand smoke (SHS) exposures are forms of exposure to tobacco smoke that are publicly well-recognized and whose negative health effects have been vigorously researched over decades. Tobacco smoke contains more than 7000 chemical components [1] that are associated with preclinical brain changes, cerebrovascular disease, cardiovascular and pulmonary disease, and developmental neurotoxicity [2]. SHS, which is generated by the combination of exhaled mainstream smoke from the direct smoker and side-stream smoke from the tobacco product itself, contains 70 carcinogenic substances [3]. Exposure of adults to SHS can seriously affect the cardiovascular system and cause heart diseases and lung cancer [4]. Compared to FHS and SHS, the concept of indirect, delayed exposure to tobacco smoke pollutants is quite new. Residual tobacco smoke, referred to as third-hand smoke (THS), occurs when tobacco smoke pollutants are absorbed, attached, or stacked on the surface of substances and dust [5]. THS remains in a re-emitted gas-phase form, oxidized form or in combination with other environmental pollutants, and plays a role as a continuous pollutant reservoir [6]. THS pollutants are involuntarily ingested, inhaled, or absorbed by the skin and the possibility of chronic exposure has also been suggested [5,7,8]. The results of experiments which exposed mice to THS (tobacco smoke-exposed cages) suggested that THS causes pathological changes in the liver, lungs, and skin [9,10]. THS exposure can also lead to impaired wound healing, lung fibrosis, hepatic steatosis, insulin resistance and diabetes, lipid abnormalities, metabolic syndrome, behavior hyperactivity, and impaired immune responses [11]. Moreover, THS pollutants interact with other environmental chemicals to produce additional pollutants. For example, nicotine, which is the most abundant pollutant in THS [3], combines with the common indoor pollutant nitrous acid (HONO) to generate carcinogenic tobacco-specific nitrosamines [8]. The health risks of THS are intensified in infants and toddlers, who commonly stay indoors for longer and are closer to the floor than are adults, increasing infant and toddler exposure to THS. Infant inhalation of house dust is double that of adults, increasing their vulnerability to THS [12]. One study including 31,584 children in South Korea showed that children exposed to THS (defined as children with smoking parents who did not smoke in presence of their child) had higher numbers of cough-related symptoms than those among children who were not exposed to THS [13]. Another paper regarding the relationship between the home-smoking rules and ETS exposure in 396 Italian children, showed that urinary cotinine level was the highest in children who had cohabitants that smoke at home even if children were on site, followed by children who had cohabitants that smoke at home when children are not on site, cohabitant smokers that never smoke at home, and non- smoking cohabitants. The authors claimed that the urinary cotinine level of children who had cohabitants that smoke at home when children are not on site or had cohabitant smokers that never smoke at home, are affected by THS from contaminated hair, clothing and from household surfaces and dust. They have also pointed out that THS is a major public health concern since it indicates the difficulty of maintaining a safe level of exposure to tobacco smoke. Authors also noted that THS is correlated with smoking habits and home-smoking precautions by cohabitants [14]. The risk of THS is further highlighted by the fact that THS is a chronic, ubiquitous, and unconscious pollutant. Compared to SHS, which is restricted to situations with direct exposure to tobacco smoke, THS pollutants may persist for more than one month depending on the degree of tobacco smoke pollutant absorption or the adhesion rates [3]. Furthermore, THS is not restricted to physical spaces in which tobacco smoke is sustained but can also be transmitted via the hands of smokers to other persons or objects [15,16]. Studies on THS and the indoor environment have generally focused on the relationship between THS and indoor furniture surfaces; used cars; and the existence of indoor smoking bans in houses, hotels, or cars. As examined by surface wipe sampling, the level of nicotine was highest in hotel rooms that permitted smoking, followed by non-smoking rooms in hotels that also had smoking rooms and smoke-free hotels [17]. The nicotine rate in the dust in smokers’ cars showed higher levels of tobacco pollutant than those of the cars of non-smokers [18,19]. THS was also shown to persist in smokers’ homes and non-smokers moving to houses formerly occupied by smokers were shown to be vulnerable to THS [16]. However, the above studies were limited to private indoor environments that are used by a limited number of people (i.e., residents of the house or owner of the car) under certain conditions (i.e., houses formally used by smokers). None of these studies assessed THS exposure in public or multi-use environments (e.g., public transportation), where the floating population is high and the environments are repeatedly used by the general public. In addition, to our knowledge, no studies have assessed the socio-economic status of populations that are especially vulnerable to THS. This study aimed to identify the public facilities and socio-demographic groups that were particularly susceptible to THS.

## 2. Materials and Methods 

### 2.1. Data Source and Study Population

Statistical analysis was performed based on the Korean National Environmental Health Survey (KNEHS) I (2009–2011). The KNEHS II (2012–2014) was not selected because it had no data on various public facilities and the KNEHS III (2015–2018) was not yet available as it was scheduled for publication in 2019. The data were collected by the National Institute of Environmental Research (NIER), which used stratified cluster sampling design based on geographic and socio-economic factors. The KNEHS I contains data from 6311 adults older than 19 years of age. All participants provided written consent to participate in the study with study approval of internal review board (NIER, Department of Environmental Health Research – 1805). Data were collected by questionnaires and blood and urine samples. The details of the KNEHS I were described in previous studies [20,21]. To exclude the effects of FHS and SHS, we restricted our respondents based on the responses to the questionnaires as follows: those who (1) never smoked or were former smokers (*n* = 3815) and (2) reported no SHS exposure (*n* = 2411). Among 2,411 respondents, those whose (1) urinary cotinine concentrations (UCCs) were not reported (*n* = 10), (2), UCCs were not estimated (*n* = 285), and (3) UCCs were high enough (>100 ng/mL) to suggest smoking (*n* = 31) were excluded. After excluding respondents with missing data on (1) whether they were living with smokers (*n* = 698), (2) household income (*n* = 24), or (3) occupation (*n* = 3), data from 1360 participants were ultimately used for further analysis.

### 2.2. Variables

The socio-demographic variables included (1) sex, (2) age group, (3) marital status, (4) level of education, (5) level of monthly income, (6) occupational group (administrative and professionals; clerical; service and sales; agricultural, forestry, and fisheries; technicians and mechanics; simple labor; housewife; student; or unemployed), (7) housing type, and (8) the existence of a smoker within the family. The levels of monthly income were reclassified into quartiles. The variables associated with the frequency of use of various public facilities included: (1) use of public transportation (yes/no), (2) type of public transportation (bus/subway/train/taxi/none), (3) average weekly frequency of public transportation use (none/1–3 times/4–7 times/8 times or more), and (4) frequency (none/sometimes (1–3 times per month]/frequently (4–7 times per week)) of public facility (theaters and auditoriums; business facilities; restaurants; sauna; academic institutes; hair-salons; internet cafes; karaoke; bars; indoor sports facilities; cultural education centers; and religious facilities) use. The average weekly frequency of public transportation use was categorized into quartiles. 

### 2.3. UCC

The effect of smoking was assessed according to UCC. UCC and urinary creatinine analyses were carried out at an analytical laboratory certified by the Korean Ministry of Health and Welfare. Urinary cotinine concentrations were measured using gas chromatograph-mass spectrometry. Urinary creatinine was determined with an alkaline picrate kinetic (Jaffe) method using an Adiva 2400 Chemistry System (Siemens Healthcare Diagnostics). The specific method of urinary cotinine analysis has been described elsewhere [20,21]. The method detection limit (MDL) for urinary cotinine was 0.27 ng/mL and concentrations below the MDL were described as 0.2 ng/mL (MDL/√2) and were included for data analysis as 0.2 ng/mL. Among our 1360 participants there were 146 participants whose urinary cotinine was described and used for data analysis as 0.2 ng/mL, so there could be minor difference on a first decimal point basis between the urinary cotinine concentration results and reality. UCCs were normalized to urinary creatinine concentrations. 

### 2.4. Statistical Analysis

IBM SPSS Statistics for Windows, version 23.0 (IBM, Armonk, NY, USA) was used for the statistical analyses. Because of the skewness of the UCC distributions, the distributions of the socio-demographic factors were described as geometric means and natural log transformations were used for the analyses when some previous studies used the Kruskal-Wallis test and Mann-Whitney tests as analytical methods [14]. *T*-test and analysis of variance (ANOVA) with Scheffe’s post-hoc analysis were used for univariate analysis. Covariates with *p* < 0.25 in univariate analysis were included in the initial generalized linear model (GLM) and covariates with *p* < 0.05 were included in the backward selection method.

## 3. Results

The geometric means (GMs) and standard errors (SE) of UCC by socio-demographic variables are shown in Table 1. Most of the participants were female (81.9%), aged between 40 and 59 years (47.9%), married (85.1%), housewives (38.2%), residents of apartments (43.2%), and had no family members that smoked (71.5%). 

The GMs and SEs of UCC by public facilities variables are shown in Table 2. Most of the participants used mass transportation (54.1%) and buses (39.6%) and used mass transportation eight times or more per week (19.9%). Among public facilities, the majority of participants rarely used theaters and auditoriums (77.9%), academic institutes (95.8%), internet cafes (98.2%), karaoke (90.1%), bars (81.3%), indoor sports facilities (86.0%), or cultural education centers (88.9%).

The socio-demographic factors that showed significant differences in log-transformed UCCs by t-test and ANOVA included sex, marital state, level of education, occupational group, housing type, and the presence of a smoker within the family (Figure 1). Among socio-demographic variables, the log UCC of women (arithmetic mean [AM]: 0.48 ng/mg Cr) was significantly higher than that in men (AM: 0.24 ng/mg Cr). Participants who had gone through divorce or bereavement (AM: 0.88 ng/mg Cr) had a significantly higher log UCC than those who were single (AM: 0.29 ng/mg Cr) and married (AM: 0.43 ng/mg Cr). Significant differences were also observed in the level of education, with participants with masters or higher degrees (AM: 0.04 ng/mg Cr) having a lower log UCC than that of those with a bachelors’ degree (AM: 0.30 ng/mg Cr), high school education (AM: 0.42 ng/mg Cr), elementary and secondary school education (AM: 0.59 ng/mg Cr), and no education (AM: 0.86 ng/mg Cr). While significant differences were observed for occupational groups, there were no differences in post-hoc analysis. The UCCs were highest for service and sales (AM: 0.73 ng/mg Cr); followed by simple labor (AM: 0.71 ng/mg Cr); housewives (AM: 0.44 ng/mg Cr); technicians and mechanics (AM: 0.40 ng/mg Cr), unemployed (AM: 0.40 ng/mg Cr), agricultural, forestry and fisheries (AM: 0.34 ng/mg Cr); clerical (AM: 0.32 ng/mg Cr); administrative or professionals (AM: 0.31 ng/mg Cr); and students (AM: 0.23 ng/mg Cr). By housing type, participants living in multiunit houses (AM: 0.77 ng/mg Cr) had the highest UCC compared to those among those living in non-residential buildings (AM: 0.26 ng/mg Cr) and apartments (AM: 0.28 ng/mg Cr). The existence of a smoking family member significantly affected the UCC (AM: 0.90 vs. 0.26 ng/mg Cr), while there were no significant differences in UCCs according to age groups and household income. 

Among public facilities, the average frequencies of weekly public transportation use, sauna use, and bar visits were associated with significant differences in log UCCs by *t*-test and ANOVA (Figure 2). The AM of the natural-log transformed UCCs of participants who used public transportations eight times or more per week (0.59 ng/mg Cr) was significantly higher than that in residents who used public transportation 1–3 times per week (0.28 ng/mg Cr). Frequent users of saunas (AM: 0.59 ng/mg Cr) had significantly higher UCCs than those who rarely used saunas (AM: 0.36 ng/mg Cr). The UCCs differed significantly between participants who rarely visited bars (AM: 0.40 ng/mg Cr), who sometimes visited bars (AM: 0.56 ng/mg Cr), and who frequently visited bars (AM: 0.74 ng/mg Cr). There were no significant differences in the UCCs of participants who used public transportation; type of public transportation; and the frequency of use of theaters and auditoriums, business facilities, restaurants, academic institutes, hair-salons, internet cafes, karaoke, indoor sports facilities, cultural education centers, and religious facilities.

The factors of public facilities that showed significant differences in log UCC by GLM included the weekly average frequency of public transportation use, and the frequencies of use of saunas, internet cafes, and bars (Table 3). The UCC of residents who used public transportations eight times or more per week (exp. (β) = 1.25) were significantly higher than that of participants who never used public transportation. In public facilities, the log UCC of participants who frequently used saunas (exp. (β) = 1.16) was significantly higher than that of participants who rarely used saunas. Frequent visitors of internet cafes (exp. (β) = 2.87) had a significantly higher UCC than that of participants who rarely used internet cafes. Frequent visitors of bars (exp. (β) = 1.59) had a significantly higher log UCC than those who did not. Among socio-demographic factors, the log UCC of participants who had experienced divorce or bereavement (exp. (β) = 1.74) or who were married (exp. (β) = 1.26) were significantly higher than that of participants who were single. The log UCCs of uneducated participants (exp. (β) = 1.99) or with elementary or secondary school education (exp. (β) = 1.52) were significantly higher than that of participants who had master’s degrees or higher. The log UCC of participants with smoking family members (exp. (β) = 1.82) was also significantly higher than that in participants without a smoking family member. 

## 4. Discussion

Our results from the GLM revealed that the frequent use of public transportation, saunas, bars, and internet cafes significantly impacted THS exposure. Participants who had experienced divorce or bereavement, had low educational levels, and had smokers within their families were also susceptible to THS.

### 4.1. Frequencies of Public Facility Use 

Our results indicate that the frequent use of public transportation may lead to higher THS exposure. Frequent users of public transportation are likely to have increased exposure to other smokers also using public transportation as well as to residual tobacco smoke pollutants carried by smokers. Given that nicotine concentration in dust (mean = 11.61 mg/g) and on surfaces (mean = 5.09 mg/m^2^) of smokers’ cars in areas with a smoking ban were significantly higher than that in the dust (mean = 3.37 mg/g) and surfaces (mean = 0.06 mg/m^2^) of non-smokers’ cars [19], contamination in public transportation is possible in the same way despite the imposition of a smoking ban. Our results also showed that the type of public transportation and the use of public transportation itself were irrelevant to THS exposure. This suggests that the repeated use of public transportation has a greater impact on THS exposure than the type or use of public transportation, which implies that the degree of consistency is a crucial factor that must be considered when studying THS exposure in public facilities.

Regarding public facilities, frequent sauna users had a significantly higher UCC due to THS. These results can be interpreted in two ways. First, the sauna itself may be heavily contaminated with THS due to the leakage of tobacco smoke from smoking rooms within the sauna. Second, the sauna may be a place where a larger area of skin is directly exposed to THS pollutants on surfaces or in the air. The interpretation in this respect requires prior understanding that one of the major accumulation pathways of THS is dermal uptake [5]. Even if the level of contamination is low, users may be vulnerable to high THS accumulation in special circumstances (i.e., highly exposed skin). This also implies that commonly used methods such as surface swipe may not be sufficient when measuring THS exposure.

Frequent internet cafe users were also vulnerable to THS exposure, showing the highest UCC among other frequent multi-use facility users. Even though the number of frequent internet cafe users was small (*n* = 9), they had a distinctively high UCC. This finding indicates the possibility that frequent internet cafe users are at high risk for THS exposure. A study conducted in South Korea showed that even after the enforcement of smoking bans in internet cafes in 2014, smokers were found in about 47% of internet cafes and that enforcement had no impact on improving the indoor air quality [22]. Our study data were collected from 2009 to 2011, before the enforcement of the indoor smoking ban in internet cafes in 2014. Therefore, we presume that more than 47% of internet cafes with smokers before the law enforcement would have contributed to the deep THS reservoir in internet cafes. Since our participants were restricted to individuals who have never smoked and had no exposure to SHS, it is highly likely that they visited internet cafes which autonomously imported smoking ban before the enforcement of legislation. Still, the tendency of high ratio of smokers visiting internet cafes (47% after the enforcement of smoking ban) and that there are many communal devices such as keyboards or mouse which especially requires the use of hands, clearly defines the high possibility of frequent exposure to THS pollutants carried by smokers. Furthermore, even if the smoking ban is enforced and that internet cafes now have separated smoking rooms, several studies have reported leakage of tobacco smoke from smoking rooms [23,24]. This indicates that internet cafes are still at risk for THS, although there may be differences in degree.

Frequent users of bars also had higher THS exposure. Studies have shown that employees of bars and nightclubs have the highest levels of SHS among public places [25]; thus, it is likely that bars also have deeper residual tobacco smoke reservoirs. Moreover, given the high possibility of SHS exposure around bars [26], tobacco smoke pollutants may also travel indoors, deepening their reservoirs.

Because we relied on the responses from the questionnaires regarding SHS exposure, we could not completely exclude the possibility of unconscious exposure of SHS, especially in facilities such as internet cafes and bars where FHS and SHS exposure could highly occur. Nevertheless, we would like to note that it such facilities have clearly high possibility of exposure to THS pollutants carried by usual visitors of smokers or leakage from outdoors around the facility even though there were no FHS or SHS exposure within the facility.

### 4.2. Socio-Demographic Factors

Previous studies on the proportions of smokers and marital status indicated that smoking rates were lower in the order of married, single, divorced, and bereaved individuals [27,28]. However, our results suggested that single participants had the lowest exposure to THS. This reversed result can be explained by the differences in the levels of autonomy. Within our data, single participants were mostly unemployed or students (44.2%), approximately five times higher than that in married participants (9.1%). Married participants were mostly housewives (43%). That singles were mostly unemployed or students indicates that they are more likely to autonomously choose their environment and companions, which also reduces their exposure to places vulnerable to THS. In contrast, married participants are more likely to be in situations, whether in occupational environments or in the case of housewives, the smoking of their spouses, where their level of control is lower than that of single individuals. While the smoking proportions of divorced or bereaved participants are well explained by the stress-relieving effects of tobacco [28], the highest levels of THS exposure among divorced or bereaved participants require further research. Within our data, the frequency with which divorced or bereaved participants visited facilities that were vulnerable to THS (i.e., public transportation, sauna, bar, internet cafe) was not particularly high compared to the frequencies in other groups.

The UCC tendencies according to education matched those for FHS and SHS, with participants with higher education levels having lower UCCs than those with less education. Participants with higher levels of education tend to have a higher awareness of the health risks of tobacco smoke [29]. This knowledge increases the likelihood of intentionally avoiding places in which smoking occurs frequently, which lowers the exposure not only to SHS but also to residual tobacco smoke pollutants. 

Participants with smoking family members showed a higher UCC than that of those without smoking family members. Regarding participants without SHS exposure, it is likely that the smoking family member did not smoke in the house or in the presence of the participant. This indicates that THS exposure could not be prevented, even with smoking-related house rules.

UCC differences by housing type had borderline significance. Multi-unit house residents, who had the highest UCC, may have increased exposure to residual tobacco smoke traveling not only through central ventilation and shared public places such as hallways [30] but also from the outdoors, since most multiunit houses have less than six stories. Furthermore, frequent turnover rates or changes of ownership in indoor environments present a high risk of THS [6]; multiunit houses may have particularly higher turnover rates compared to those of other housing types. However, to our knowledge, differences in turnover rates by housing type have not yet been reported in Korea. Participants living in apartments had a low UCC. Given that residents living in apartments generally have higher social economic positions in South Korea [31] and the prevalence of smoking is lower in people with higher social economic positions [32] it is likely that participants living in apartments had fewer smokers in their neighborhoods. 

Although sex and occupation did not show significant differences in the GLM, it is worthwhile to discuss the results of the univariate analysis. The results showed significantly higher UCC in women than those in men; however, most studies on active or passive smoking have reported higher levels in men, although the gap is declining [33,34]. This difference might be due to the proportion of no-FSH and no-SHS in each sex. Within our study restrictions, men who had never smoked and who reported having no SHS exposure were likely a minor proportion of men, whereas the same conditions among women are not as uncommon, given the higher smoking rate and SHS exposure in men than those in women [33]. Our own results also support the presumption that men who were not current smokers and who had no exposure to SHS comprised a lower percentage (11.8%) of the male population than the comparable percentage among women (32.1%). The occupational tendency observed for UCC mostly matched the level of education. Occupations that did not require high levels of education, such as simple labor, had a higher UCC (GM: 2.04 ng/mg Cr), that that for occupations that required higher levels of education, such as administrative or professionals (GM: 1.37 ng/mg Cr). Furthermore, most of our results were consistent with the smoking prevalence by occupation reported in the United States [18,35], in which simple laborers and construction workers had high UCCs and professionals or managers had low UCCs. However, the distinctively high UCCs of participants in sales and service did not match the prevalence of smokers by occupation. Even though our study excluded participants exposed to FHS or SHS, the concordance of the majority of our results to the smoking prevalence by occupation suggests a high possibility for exposure to residual smoke pollutants even without smoking and exposure to SHS when there is a high rate of smokers in the occupational environment, which lead to the accumulation of UCC in non-smokers. In addition, the UCCs of sales and service occupations suggest that even though they have fewer smoking colleagues compared to those of other occupations, occupations that require meeting with numerous people with various backgrounds in various environments have a higher possibility of THS exposure.

It should be noted that the group of individuals who are not exposed to FHS and SHS may have a specific distribution within socio-demographic factors. Within the total data of the KNEHS I, the distribution patterns of current smokers for education (masters or higher: 2.5%, bachelors: 34.7%, high school: 40.7%, elementary and secondary: 20.8%, none: 1.3%), sex (male: 91.4%, female: 8.6%) differed from those of our study population (THS exposure only), while marital state (single: 20.8%, married: 76.0%, divorced or bereavement: 3.7%) showed a similar distribution pattern. Our study results of marital status, education, and sex should be considered to be suggestions that these populations may be vulnerable to THS exposure under the conditions of no FHS and SHS and that the total THS exposure in the whole population (i.e., those exposed to THS as well as FHS and SHS) would have a different pattern with our results. That THS health threat and degree of exposure by public facilities may differ by countries should be also considered. Countries that have a high percentage of smokers within the population, low public recognition of the health risk of FHS and SHS, non- existence of smoking ban in public facilities, and low respect of enforcement of the legislation, should consider THS exposure in public facilities more critically.

### 4.3. Limitations

Because exposure variables were collected based on the questionnaire responses, the measurements of exposure might not be correct or accurate. Although participants reported that they were not exposed to SHS, unrecognized SHS exposure was possible. While the frequency of use of public facilities was investigated, information on the time of stay was not collected. Therefore, in this study, frequent users of public facilities were presumed to spend more time in those facilities. However, the purposes of the KNEHS were not directly related to smoking problems; thus, the possibility of linkage between each participant’s UCC and response to the exposure variables was low. Hence those possible errors might not lead to bias. As we limited our study population to those who were not current smokers and were not exposed to SHS, the results and interpretation of some socio-demographic factors may vary from those obtained when data from FHS and SHS-exposed populations are included. Therefore, our results might be limited to certain populations. The small sample size of some factors may have decreased the statistical power to examine the effects. As our data were limited to adults, the results cannot explain the main exposure routes for infants. It was also hard to expand our discussion to possible preventive policies for vulnerable public facilities or populations. With recent studies suggesting e-cigarettes (e-cigs) produce airborne particulate matter (PM) and cause passive smoking dose [36] and that regardless of the smoking devices (traditional cigarettes or e-cigs) infants showed the highest doses of particles deposited in the respiratory systems, further studies should also consider e-cigs in depth [37].

### 4.4. Study Strengths

To our knowledge, this is one of very few attempts to assess THS exposure in a variety of environmental and socio-demographic conditions. Identifying indoor environments and socio-demographic groups that are particularly exposed to THS might not only raise public awareness of THS exposure but may also help the public to reduce THS exposure in certain facilities through various methods, including cleaning and vacuuming. The study results suggest the need and direction for further THS research on each factor. This study is also one of the very few studies on people who did not smoke and who also had limited exposure to SHS. Representative sampling of a nation might increase the generalizability of these findings. 

## 5. Conclusion

Among those without FHS or SHS exposure, frequent use of public transportation, saunas, internet cafes, and bars significantly impacted THS exposure. Participants with smokers within their families, who had experienced divorce or bereavement, and with low levels of education were vulnerable to THS. Possible links between housing type, sex, and occupation and THS exposure are also suggested. For improved understanding of exposure routes, further studies on each factor with real-life observation and experiments are required. Further research on THS and in additional public facilities, such as those for infants and toddlers, is also required. 

## Figures and Tables

**Figure 1 ijerph-16-00855-f001:**
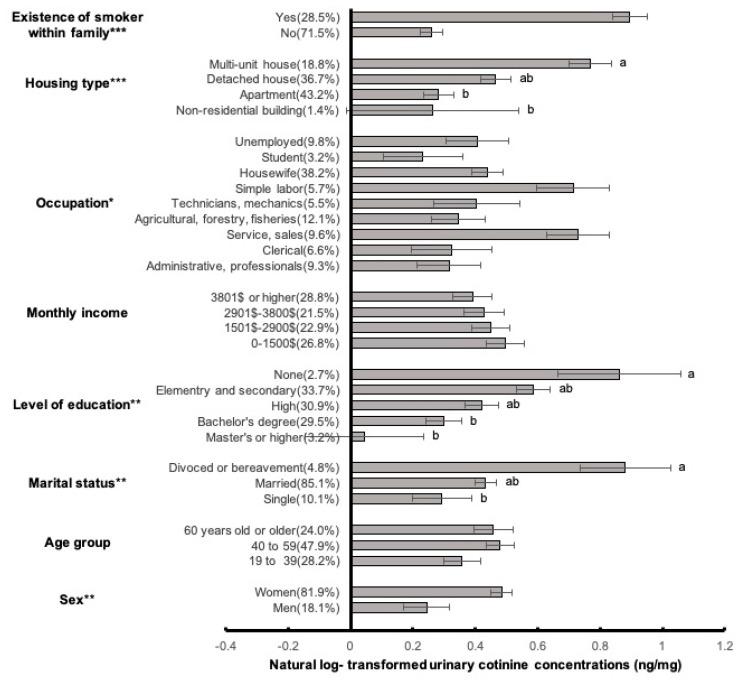
Means and standard errors of log-transformed urinary cotinine concentrations according to socio-demographic factors including sex, age group, marital status, level of education, monthly income, occupation, housing type, and the presence of a smoker within the family. *T*-test and ANOVA with Scheffe’s post-hoc test (a, b). * *p* < 0.05; ** *p* < 0.01; *** *p* < 0.001.

**Figure 2 ijerph-16-00855-f002:**
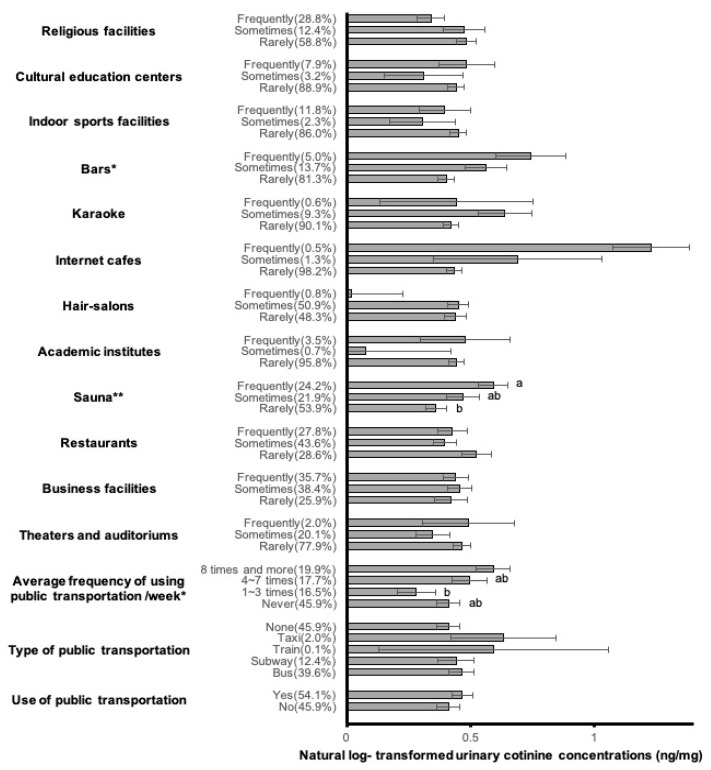
Means and standard errors of log-transformed urinary cotinine concentrations according to public facility usage including use of public transportation, type of public transportation, average weekly frequency of public transportation use, theaters and auditoriums, business facilities, restaurants, saunas, academic institutes, hair-salons, internet cafes, karaoke, bars, indoor sports facilities, cultural education centers, and religious facilities. *T*-test and ANOVA with Scheffe’s post-hoc test (a, b, c, d). * *p* < 0.05; ** *p* < 0.01; *** *p* < 0.001.

**Table 1 ijerph-16-00855-t001:** Distributions of socio-demographic factors and their associations with creatinine-adjusted urinary cotinine concentrations (ng/mg Cr).

Socio-Demographics	*n* (%)	GM (±SD)
Total	1360 (100%)	1.55 (±6.57)
Sex		
Male	246 (18.1%)	1.28 (±4.90)
Female	1114 (81.9%)	1.62 (±6.88)
Age group		
19 to 39	383 (28.2%)	1.43 (±5.44)
40 to 59	651 (47.9%)	1.62 (±7.33)
60 years old or older	326 (24.0%)	1.58 (±6.17)
Marital status		
Single	138 (10.1%)	1.34 (±4.05)
Married	1157 (85.1%)	1.54 (±6.83)
Divorced or bereaved	65 (4.8%)	2.41 (±5.99)
Level of education		
Masters or higher	44 (3.2%)	1.04 (±5.52)
Bachelor’s degree	401 (29.5%)	1.35 (±5.79)
High school	420 (30.9%)	1.52 (±4.74)
Elementary and secondary	458 (33.7%)	1.80 (±8.42)
None	37 (2.7%)	2.37 (±6.99)
Monthly income		
$0~$1500	364 (26.8%)	1.64 (±4.81)
$1501~$2900	312 (22.9%)	1.57 (±4.94)
$2901~$3800	292 (21.5%)	1.53 (±10.24)
$3801 or higher	392 (28.8%)	1.48 (±5.56)
Occupation		
Student	43 (3.2%)	1.26 (±1.64)
Administrative or professionals	127 (9.3%)	1.37 (±4.60)
Clerical	90 (6.6%)	1.38 (±6.97)
Agricultural, forestry, or fisheries	165 (12.1%)	1.41 (±4.18)
Technicians or mechanics	75 (5.5%)	1.50 (±5.05)
Unemployed	133 (9.8%)	1.50 (±3.37)
Housewife	520 (38.2%)	1.55 (±7.96)
Simple labor	77 (5.7%)	2.04 (±5.39)
Service and sales	130 (9.6%)	2.07 (±8.73)
Housing type		
Non-residential building	19 (1.4%)	1.30 (±1.93)
Apartment	587 (43.2%)	1.33 (±5.59)
Detached house	499 (36.7%)	1.59 (±7.78)
Multiunit house	255 (18.8%)	2.16 (±6.04)
Existence of smoker within family		
No	973 (71.5%)	1.30 (±6.32)
Yes	387 (28.5%)	2.45 (±6.96)

**Table 2 ijerph-16-00855-t002:** Distributions of factors related to the usage of public facilities and their associations with creatinine-adjusted urinary cotinine concentrations (ng/mg Cr).

Public Facilities	*n* (%)	GM (±SD)
Use of public transportation		
No	624 (45.9%)	1.51 (±6.30)
Yes	736 (54.1%)	1.59 (±6.80)
Type of public transportation		
Bus	539 (39.6%)	1.59 (±7.59)
Subway	168 (12.4%)	1.55 (±3.84)
Train	2 (0.1%)	1.81 (±1.23)
Taxi	27 (2.0%)	1.88 (±3.78)
None	624 (45.9%)	1.51 (±6.30)
Average frequency of using public transportation per week		
Never	624 (45.9%)	1.51 (±6.30)
1~3 times	225 (16.5%)	1.32 (±3.88)
4~7 times	241 (17.7%)	1.64 (±8.58)
8 times and more	270 (19.9%)	1.81 (±6.88)
Theaters and auditoriums		
Rarely	1060 (77.9%)	1.59 (±6.91)
Sometimes	273 (20.1%)	1.41 (±5.40)
Frequently	27 (2.0%)	1.63 (±2.41)
Business facilities		
Rarely	352 (25.9%)	1.52 (±7.42)
Sometimes	522 (38.4%)	1.58 (±4.89)
Frequently	486 (35.7%)	1.55 (±7.44)
Restaurants		
Rarely	389 (28.6%)	1.69 (±6.80)
Sometimes	593 (43.6%)	1.48 (±5.31)
Frequently	378 (27.8%)	1.53 (±7.97)
Sauna		
Rarely	733 (53.9%)	1.43 (±6.90)
Sometimes	298 (21.9%)	1.60 (±5.84)
Frequently	329 (24.2%)	1.81 (±6.46)
Academic institutes		
Rarely	1303 (95.8%)	1.55 (±5.80)
Sometimes	9 (0.7%)	1.08 (±2.06)
Frequently	48 (3.5%)	1.61 (±17.64)
Hair-salons		
Rarely	657 (48.3%)	1.55 (±7.23)
Sometimes	692 (50.9%)	1.57 (±5.93)
Frequently	11 (0.8%)	1.02 (±0.76)
Internet cafes		
Rarely	1336 (98.2%)	1.54 (±6.60)
Sometimes	17 (1.3%)	1.99 (±5.18)
Frequently	7 (0.5%)	3.41 (±1.44)
Karaoke		
Rarely	1226 (90.1%)	1.52 (±5.69)
Sometimes	126 (9.3%)	1.89 (±12.26)
Frequently	8 (0.6%)	1.55 (±1.86)
Bars		
Rarely	1106 (81.3%)	1.49 (±6.31)
Sometimes	186 (13.7%)	1.75 (±7.26)
Frequently	68 (5.0%)	2.10 (±8.36)
Indoor sports facilities		
Rarely	1169 (86.0%)	1.57 (±5.73)
Sometimes	31 (2.3%)	1.36 (±1.25)
Frequently	160 (11.8%)	1.48 (±11.24)
Cultural education centers		
Rarely	1209 (88.9%)	1.55 (±6.42)
Sometimes	44 (3.2%)	1.36 (±4.42)
Frequently	107 (7.9%)	1.62 (±8.70)
Religious facilities		
Rarely	799 (58.8%)	1.62 (±7.11)
Sometimes	169 (12.4%)	1.61 (±8.29)
Frequently	392 (28.8%)	1.40 (±4.10)

**Table 3 ijerph-16-00855-t003:** Public facility usage and socio-demographic factors related to log-urinary cotinine concentrations in generalized linear regression.

Covariates	Exp. (β)	Exp (SE)	Exp CI (95%)	*p*-Value
Average frequency of using public transportation/week				
Never	1.00			
1–3 times	0.89	0.08	(0.75, 1.05)	0.164
4–7 times	1.07	0.08	(0.91, 1.26)	0.419
8 times and more	1.25	0.08	(1.07, 1.47)	0.006
Sauna				
Rarely	1.00			
Sometimes (1–3 times per month)	1.09	0.07	(0.95, 1.27)	0.231
Frequently (1–7 times per week)	1.16	0.07	(1.00, 1.34)	0.043
Internet cafes				
Rarely	1.00			
Sometimes (1–3 times per month)	1.29	0.28	(0.75, 2.21)	0.358
Frequently (1–7 times per week)	2.87	0.42	(1.26, 6.56)	0.012
Bar				
Rarely	1.00			
Sometimes (1–3 times per month)	1.23	0.09	(1.03, 1.47)	0.02
Frequently (1–7 times per week)	1.59	0.14	(1.21, 2.10)	0.001
Marital status				
Single	1.00			
Married	1.26	0.12	(1.00, 1.58)	0.048
Divorced or bereavement	1.74	0.18	(1.22, 2.47)	0.002
Level of education				
Masters or higher	1.00			
Bachelor’s degree	1.14	0.17	(0.81, 1.60)	0.451
High school	1.28	0.17	(0.91, 1.79)	0.154
Elementary and secondary	1.52	0.17	(1.08, 2.13)	0.016
None	1.99	0.24	(1.24, 3.21)	0.005
Existence of smoker within family				
No	1.00			
Yes	1.82	0.07	(1.60, 2.07)	0

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
