# Peer review of "Public Facility Utility and Third-Hand Smoking Exposure without First and Second-Hand Smoking According to Urinary Cotinine Level"

_ijerph, 2019, doi:10.3390/ijerph16050855_

Round 1

Reviewer 1 Report

The paper is quite good and interesting even if the cotinine issue is well-known and well-studied. This is the only remark regarding this paper: the authors did not analyze the literature in deep way, they did not consider paper regarding the cotinine determination by Manigrasso et al. and Protano et al. In these papers the cotinine has been analyzed, the different analytical methods investigated and the exposure levels compared in different population. Please, cite them. Further, the approach shown by authors is in agreement with the scope of the paper, I suggest to give some information about the third-hand smoke. Finally, the authors should introduce in the conclusion the aspects related to the e-cig scenario, still from the same authors.

Author Response

We appreciate the valuable comment from you. Our response to your comment is listed below.

Point 1: they did not consider paper regarding the cotinine determination by Manigrasso et al. and Protano et al. In these papers the cotinine has been analysed, the different analytical methods investigated and the exposure levels compared in different population. Please, cite them.

Response 1:

We added the Protano et al.'s paper [1] that studied the correlation of home-smoking rules or smoking habits of cohabitant and ETS exposure on 396 Italian children (5-11 years old). Cohabitant's (or parents in most cases) smoking habits were collected by questionnaires and urinary cotinine was used as an exposure assessment measurement. Kruskal-Wallis test and Mann-Whitney tests were used as an analytical method because their urinary cotinine results were not normally distributed. This was added on our method section (page 6, line 19) along with our more detailed description of cotinine analysis (page 6, line 5).

Point 2: , the approach shown by authors is in agreement with the scope of the paper, I suggest to give some information about the third-hand smoke.

Response 2:

Protano et al's paper [1] regarding the relationship between the home-smoking rules and ETS exposure on 396 Italian children, showed that urinary cotinine level was the highest in children who had cohabitants that smoke at home even if children were on site, followed by children who had cohabitants that smokes at home when children are not on site, cohabitant smoker that never smokes at home, and non- smoking cohabitants. The authors claimed that urinary cotinine level of children who had cohabitants that smokes at home when children are not on site or had cohabitant smoker that never smokes a home, are affected by THS from contaminated hair, clothing and from household surfaces and dust. They have also pointed out that THS is a major public health concern since it signifies the difficulty to maintain a safe level of exposure to tobacco smoke. This authors also noted that THS is correlated with smoking habits and home-smoking precautions by cohabitants. We have added this information on our introduction section (page 3, line 27).

Point 3: Finally, the authors should introduce in the conclusion the aspects related to the e-cig scenario, still from the same authors.

Response 3:

That further studies should consider in depth with e-cigarettes (e-cigs) were added in the discussion section (page 17, line 25) regarding its proven relationship with SHS and its highly possible linkage between THS.  We added the study results from Protano et al's paper that evaluated the levels of airborne particulate matter (PM) emitted during the use of e-cigs, which showed that all tested e-cigs devices produced PM during their use and all models were shown to cause passive smoking dose [2]. We have also noted from Protano and Manigrasso's paper that regardless of the smoking devices, the highest doses of particles deposited in the respiratory systems were shown in infants from the paper that compared the SHS generated by traditional cigarettes and electronic smoking devices with age-related dose assessment [3].

1.         Protano, C.; Andreoli, R.; Manini, P.; Vitali, M. How home-smoking habits affect children: a cross-sectional study using urinary cotinine measurement in Italy. Int. J. Public Health 2012, 57, 885-892, doi:10.1007/s00038-012-0354-0.

2.         Protano, C.; Avino, P.; Manigrasso, M.; Vivaldi, V.; Perna, F.; Valeriani, F.; Vitali, M. Environmental Electronic Vape Exposure from Four Different Generations of Electronic Cigarettes: Airborne Particulate Matter Levels. Int. J. Environ. Res. Public. Health 2018, 15, doi:10.3390/ijerph15102172.

3.         Protano, C.; Manigrasso, M.; Avino, P.; Vitali, M. Second-hand smoke generated by combustion and electronic smoking devices used in real scenarios: Ultrafine particle pollution and age-related dose assessment. Environ. Int. 2017, 107, 190-195, doi:10.1016/j.envint.2017.07.014.

Reviewer 2 Report

The manuscript is well organized, results are clearly presented and references are up-to-date.

Minor comments:

1. Although the method for determining cotinine was described in previous publications, I would suggest to add the main characteristics of the method (type of extraction, type of instrument, LOD).

2. What about participant with urinary cotinine below LOD?

3. If your study data were collected from 2009 to 2011, before the enforcement of the indoor smoking bans in internet cafes, bars and similar facilities in 2014, how is it possible that your participants were not exposed to SHS in these facilities?

Author Response

We appreciate the valuable comment from you. Our response to your comment is listed below.

Point 1:  Although the method for determining cotinine was described in previous publications, I would suggest to add the main characteristics of the method (type of extraction, type of instrument, LOD).

Response 1:

All urinary cotinine analyses were carried out at analytical laboratory certified by the Korean Ministry of Health and Welfare. Urinary cotinine concentrations were measured using gas chromatograph-mass spectrometry. Urinary creatinine was determined with an alkaline picrate kinetic (Jaffe) method using an Adiva 2400 Chemistry System (Siemens Healthcare Diagnostics). This was added on our manuscript in the method section (page 6, line 5)

Point 2: What about participant with urinary cotinine below LOD?

Response 2:

The method detection limit (MDL) for urinary cotinine was 0.27 ng/ml and concentrations below the MDL were described as 0.2 ng/ml (MDL/Ö2) and were included for data analysis as 0.2 ng/ml. Among our 1360 participants there were 146 participants' urinary cotinine that were described and used for data analysis as 0.2 ng/ml, so there could be minor difference in first decimal point basis in urinary cotinine concentration results with the reality. This was added on our manuscript in the method section (page 6, line 9)

Point 3: If your study data were collected from 2009 to 2011, before the enforcement of the indoor smoking bans in internet cafes, bars and similar facilities in 2014, how is it possible that your participants were not exposed to SHS in these facilities?

Response 3:

  Even though the law enforcement of smoking ban in such facilities was imposed in 2014, there were internet cafes and bars with smoking ban before the enforcement for reasons such as tobacco odor or the prevention of fire. Since our participants were restricted to ones who have never smoked and had no exposure to SHS, there are high possibility that they visited facilities with smoking ban. Still, we brought the issue of the percentage of smokers in internet cafe after the law enforcement and the high level of SHS exposure in employees of bar to explain that such places have high possibility of frequent exposure to THS pollutants carried by smokers or leakage from outdoors around the facility even though there were no FHS or SHS exposure within the facility. However, as you pointed out and we stated in our limitation section, because we relied on the questionnaire responses, there are possibilities of unrecognized SHS exposure of the participants.  The above were added after the discussion section of internet cafes and bars. (page 14, line 6; page 14, line 19).

Reviewer 3 Report

This paper presents a very original research, as very few papers and only very recently deal with the problem of third hand smoke. The paper is well written and the experiments well conducted. The limitations of the study are clearly stated by the authors themselves.

It is difficult to distinguish between third and second hand smoke exposure based only on urinary cotonine and a questionnaire. This in particularly evident in the paragraph about internet cafe (4.1).

This point should be mentioned other than in the limitation paragraph.

Besides this problem is more important in some countries than in others, depending on the amount of smokers in the population, on the existence of smoking bans in public places and on how much bans are respected.

This  point also should be mentioned.

Author Response

We appreciate the valuable comment from you. Our response to your comment is listed below.

Point 1:  It is difficult to distinguish between third and second hand smoke exposure based only on urinary cotinine and a questionnaire. This in particularly evident in the paragraph about internet cafe (4.1). This point should be mentioned other than in the limitation paragraph.

Response 1:

As you pointed out, because we relied on the responses from the questionnaires regarding SHS exposure, we could not exclude the possibility of unconscious exposure of SHS. Nevertheless, even if there were unconscious SHS exposure within the facility, that there is high ratio of smokers in internet cafes and that there are many communal devices such as keyboards or mouse which especially requires the use of hands, still clearly defines the high possibility of frequent exposure to THS pollutants carried by smokers. We added this point after discussing internet cafe (page 14, line 6).

Point 2: Besides this problem is more important in some countries than in others, depending on the amount of smokers in the population, on the existence of smoking bans in public places and on how much bans are respected.

Response 2:

We added the possibility of difference in degree in THS health threat by countries in a separate paragraph after discussing internet cafes and bars.  In this paragraph, we also addressed that in some countries, especially when there are high percentage of smokers within the population; low public recognition of the health risk of FHS and SHS; non-existence of smoking ban in public facilities; and low respect or enforcement of the legislation, THS exposure in public facilities should be more crucially considered (page 17, line 5).